# Effect of Baicalin on Wound Healing in a Mouse Model of Pressure Ulcers

**DOI:** 10.3390/ijms24010329

**Published:** 2022-12-25

**Authors:** Eunbin Kim, Seoyoon Ham, Bok Ki Jung, Jin-Woo Park, Jihee Kim, Ju Hee Lee

**Affiliations:** 1Department of Dermatology & Cutaneous Biology Research Institute, Yonsei University College of Medicine, Seoul 03722, Republic of Korea; 2Department of Materials Science and Engineering, Yonsei University, Seoul 03722, Republic of Korea; 3Department of Plastics and Reconstructive Surgery, Yongin Severance Hospital, Yongin 16995, Republic of Korea; 4Scar Laser and Plastic Surgery Center, Yonsei Cancer Hospital, Seoul 03722, Republic of Korea; 5Department of Dermatology, Yongin Severance Hospital, Yongin 16995, Republic of Korea

**Keywords:** pressure ulcers, flavonoid, baicalin, wound healing

## Abstract

One of the most frequent comorbidities that develop in chronically ill or immobilized patients is pressure ulcers, also known as bed sores. Despite ischemia-reperfusion (I/R)-induced skin lesion having been identified as a primary cause of pressure ulcers, wound management efforts have so far failed to significantly improve outcomes. Baicalin, or 5,6,7-trihydroxyflavone, is a type of flavonoid which has been shown to possess a variety of biological characteristics, including antioxidative and anti-inflammatory effects and protection of I/R injury. In vitro wound scratch assay was first used to assess the function of baicalin in wound healing. We established a mouse model of advanced stage pressure ulcers with repeated cycles of I/R pressure load. In this model, topically applied baicalin (100 mg/mL) induced a significant increase in the wound healing process measured by wound area. Histological examination of the pressure ulcer mouse model showed faster granulation tissue formation and re-epithelization in the baicalin-treated group. Next, baicalin downregulated pro-inflammatory cytokines (IL-6 and IL-1β), while upregulating the anti-inflammatory IL-10. Additionally, baicalin induced an increase in several growth factors (VEGF, FGF-2, PDGF-β, and CTGF), promoting the wound healing process. Our results suggest that baicalin could serve as a promising agent for the treatment of pressures ulcers.

## 1. Introduction

Pressure ulcers are a common skin condition that frequently affects elderly people and patients who are bedridden and physically impaired. Pressure ulcers are brought on by prolonged pressure applied to a specific area of tissue, which causes ischemia in the skin, subcutaneous tissue, and muscles close to the wound [1]. When pressure is constantly applied to one part of the body, typically the buttocks, hips, and heels, blood circulation is disrupted, resulting in an insufficient supply of oxygen and nutrients. Pressure ulcers are the visible evidence of pathological changes in the blood supply to dermal tissues [2]. Because of how subtle they are, early-stage pressure ulcers are difficult for non-medical staff to identify visually. The repeatedly affected area develops ulcers as the lesion progresses, causing epithelial injury and erosion at the site. In some cases, surgery can be considered for patients who exhibit medical comorbidities that may result in poor wound healing. Therefore, clinically relevant biosignals that proactively detect decubitus ulcers in early-stage pressure ulcers using timely information can be beneficial [3,4,5]. However, advanced-stage ulcers remain clinically challenging and require surgical intervention, resulting in significant patient morbidity and medical expenses [6].

The activity of cytokines and growth factors regulates the complex process of wound healing, which has several overlapping stages. In chronic wounds, the expression of inflammatory cytokines is upregulated, and the wound microenvironment sequesters different growth factors and cytokines to inhibit their functions, delaying wound healing [7]. To promote wound healing effectively, various nutraceuticals have been studied as potential treatments for chronic wounds such as pressure ulcers [8,9,10]. Recently, an increasing number of studies have involved the development of novel biomaterials with various active ingredients for the prevention of pressure ulcers [11,12,13]. Among them, flavonoids containing baicalin have demonstrated anti-inflammatory, anti-oxidative, and anti-proliferative effects [14,15,16,17,18]. Baicalin (5,6-dihydroxy-2-phenyl-4H-1-benzopyran-4-one-7-O-D-b-glucuronic acid) is a plant-derived flavonoid isolated from *Scutellaria baicalensis Georgi*. Flavonoids have been used to treat pressure and diabetic ulcers [10,11,19,20,21,22], but few studies have investigated the effect of the flavonoid baicalin on pressure ulcers, and its underlying molecular mechanisms remain to be clearly defined. Therefore, this study aimed to explore the effects of baicalin on wound healing in a mouse model of pressure ulcers. We also evaluated the therapeutic benefits of the anti-inflammatory activity of baicalin.

## 2. Results

### 2.1. Effect of Baicalin in Human Epidermal Keratinocytes (HEKs)

To evaluate the effects of wound closure in HEKs treated with baicalin, scratch assays of cell migration were performed. After scratching, HEKs were treated with baicalin (100 μg/mL) and monitored at 6, 24, and 48 h. A Cell Counting Kit-8 (CCK-8) assay was performed to measure the cytotoxicity of baicalin, and results of the CCK-8 are shown Appendix A. Wound closure was promoted in cells treated with baicalin compared with that in the control group (Figure 1A). In particular, the scratch area was reduced significantly at 48 h in HEKs treated with baicalin (** *p* < 0.01, *** *p* < 0.001; Figure 1B). Thus, baicalin enhances wound healing as cell proliferation and migration are stimulated in HEKs treated with baicalin.

### 2.2. Topical Application of Baicalin Accelerates Wound Healing in Mice with Pressure Ulcers

To confirm the wound healing activity of baicalin in mice with pressure ulcers, we compared the control and baicalin groups after skin lesions were induced in the mice. The ischemia-reperfusion (I/R) pressure mouse models revealed that the experimental group had decreased wound areas over time (Figure 2A). The wound closure rate of each site was expressed as a percentage of the initial wound area by measuring the area of the remaining wound site at 1-day intervals. Wound size in the experimental groups was measured 1, 3, 7, and 10 days after wound induction. The wound recovery speed was significantly faster in the baicalin-treated group than in the control group (* *p* < 0.05, ** *p* < 0.01; Figure 2B).

### 2.3. Treatment with Baicalin Accelerates the Restoration of Skin Structures in Mice with Pressure Ulcers

To evaluate whether baicalin can accelerate the recovery of skin structures in a murine model of pressure ulcers, skin samples from each group were collected 1, 3, 7, and 10 days after treatment and stained with hematoxylin and eosin (H&E). Histopathological analysis revealed that the baicalin treatment group had improved epithelial thickness and structure compared to the control group. On the 10th day, the baicalin group showed nearly complete epithelial regeneration with normal keratinization and remodeling of connective tissue (Figure 3). In particular, on the third day, inflammatory cells such as macrophages and neutrophils were detected in subcutaneous tissue as well as ordered connective tissue containing reduced inflammatory cells in the baicalin-treated group. Additionally, Masson’s trichrome staining revealed the organization of collagen fibers and granulation tissue formation (Figure 4A). Collagen fibers were organized and oriented in wound granulation sites and the subcutaneous tissue promoted regeneration in the baicalin-treated group. Additionally, the re-epithelialization period was short in the baicalin-treated group and prolonged in the control group. Treatment with baicalin accelerated re-epithelialization that was not only associated with an increase in epidermal thickness, but also promoted the thickness of the granulation tissue (Figure 4B). These results indicate that baicalin treatment promotes the wound healing process in a pressure ulcer mouse model.

### 2.4. Effect of Baicalin on Keratinocyte Migration in Pressure Ulcer Wound Healing

To examine the effects of baicalin on keratinocyte migration, we evaluated cytokeratin 17 (CK17), which is expressed by activated keratinocytes, via immunofluorescence. Topical application of baicalin markedly increased CK17 expression in keratinocytes as well as accelerated re-epithelialization compared to the control group from day 3 to day 7 (Figure 5). The effect was greatest on day 7. Thus, treatment with baicalin is effective in wound healing as indicated by increased CK17 expression.

### 2.5. Baicalin Alters Pro-Inflammatory and Anti-Inflammatory Cytokine Gene Expression in Mice with Pressure Ulcers

Pressure ulcers are exacerbated by chronic inflammation due to the overproduction of inflammatory cytokines. Using quantitative real-time PCR (qRT-PCR), we investigated the effect of baicalin treatment on the expression of the pro-inflammatory cytokines IL-6 and IL-1β, key factors in inflammation, after ischemia-reperfusion (I/R). On day 3 post-I/R, the mRNA levels of IL-6 (Figure 6A) and IL-1β (Figure 6B) were significantly reduced in the baicalin-treated group, with IL-1β mRNA expression significantly reduced on days 1 and 3 post-I/R. In addition, the mRNA levels of the anti-inflammatory cytokines IL-10 and TGF-β under the I/R condition were increased in the baicalin group (Figure 6C,D). These results suggest that baicalin suppresses pro-inflammatory cytokine expression and induces anti-inflammatory cytokine expression to regulate the inflammatory response in a pressure ulcer mouse model.

### 2.6. Effect of Baicalin on Angiogenesis-Related Gene Expression in Mice with Pressure Ulcers

To verify the angiogenic activity of baicalin, we conducted qRT-PCR to assess mRNA expression. qRT-PCR analysis revealed that the expression levels of VEGF, FGF-2, and PDGF-β, proangiogenic markers, were significantly elevated (*p* < 0.05) in the baicalin group compared to the control group on day 3 post-I/R (Figure 7). CTGF expression was not significantly altered by baicalin treatment. Therefore, baicalin stimulation of wound angiogenesis may be mediated by modulation of angiogenesis-related gene expression.

## 3. Discussion

In the present study, we investigated the effect of baicalin on wound healing in a murine model of advanced-stage pressure ulcers. Baicalin induced faster re-epithelization and granulation tissue formation and downregulated pro-inflammatory factors, suggesting its anti-inflammatory potential during the wound healing process. In addition, baicalin stimulated angiogenesis, which can alter the expression of cytokines and growth factors.

Baicalin is a flavonoid compound extracted from the root of *S. baicalensis* Georgi. Baicalin exhibits various pharmacological activities, including anti-inflammatory, antioxidant, and anti-allergic therapies. Previous studies have demonstrated that baicalin protects against pressure stress such as that found in type-2 diabetes and cerebral ischemia injury [23,24,25,26]. Our study suggests that baicalin can reduce advanced-stage pressure ulcers by promoting wound healing related to anti-inflammatory activity. By histological analysis, advanced pressure ulcers of mice treated with baicalin demonstrated accelerated re-epithelialization and formation of granulation tissue compared to untreated ulcers.

Cytokeratin 17 (CK17) is a basal/myoepithelial cell-associated keratin protein expressed in the outer root sheath of wild-type hair follicles [27]. It is not expressed in the normal epidermis, but CK17 expression is induced in activated keratinocytes [28]. In the present study, immunofluorescence staining showed significant increases in CK17 expression following baicalin treatment compared to that in the control group. Therefore, baicalin may play a role in promoting keratinocyte migration, thus promoting wound healing in chronic pressure stress ulcers. Additionally, topical application of baicalin on reconstituted hair follicles in the mice induced increased development of the terminal hair by activating Wnt/β-catenin signaling and dermal papilla cells [29].

During the wound healing process, various cytokines and growth factors are released simultaneously and continuously in the wound area. Their ability to maximize regenerative advantages and minimize side effects can help accelerate wound healing and promote tissue regeneration. Pro-inflammatory cytokines such as IL-6 and IL-1β play crucial roles in wound healing by impacting various processes at the wound site, including keratinocyte and fibroblast proliferation, synthesis and breakdown of extracellular matrix proteins, fibroblast chemotaxis, and immune response regulation [30,31,32]. IL-10 and TFG-β suppress further cytokine production, resulting in anti-inflammatory wound healing effects [33,34].

We evaluated cytokine gene expression using qRT-PCR. IL-6 and IL-1β mRNA levels were downregulated following baicalin treatment starting from day 1, suggesting early resolution of the inflammatory phase of wound healing, whereas IL-10 and TFG-β mRNA levels were upregulated in chronic pressure ulcer wounds compared with those in normal skin. Based on the well-characterized roles of growth factors in wound healing, we assessed the expression of the wound healing factors VEGF, FGF-2, CTGF, and PDGF-β as they play crucial roles in the angiogenesis process. VEGF induces the re-programming of monocytes to become angiogenic [35]. CTGF promotes endothelial proliferation, migration, and adhesion in angiogenesis. FGF-2 plays a critical role in granular tissue formation, re-epithelialization, and tissue remodeling [36,37]. In a clinical study, FGF-2-treated pressure ulcer patients experienced faster wound closure. PDGF-β is essential for vascular formation and can enhance the survival, proliferation, and migration of endothelial cells, further promoting angiogenesis. PDGF-β expression can be induced by hypoxia, which is a critical stimulus of the wound healing process [38]. The mRNA expression of these growth factors was significantly elevated in the skin tissue samples from the baicalin-treated group after ischemia-reperfusion (I/R) compared to the untreated group. Upregulation of these growth factors can enhance the healing of pressure ulcers. We also confirmed the protein level by examining ERK phosphorylation to analyze baicalin's effect in a pressure ulcer mouse model. There was a significant increase in ERK phosphorylation on day 1 after the application of baicalin (Appendix A). Such findings are consistent with findings observed in RT-PCR studies showing the resolution of inflammation.

Overall, our data suggest that the application of baicalin may modulate inflammation in advanced stage pressure ulcers by decreasing the early pro-inflammatory cytokine. Additionally, our results demonstrated that baicalin stimulates re-epithelization and granulation tissue formation in the wound healing process of advanced pressure ulcers.

## 4. Materials and Methods

### 4.1. Cell Culture

Human epidermal keratinocytes (HEKs) were obtained from ATCC (PCS-200-010). HEKs were cultured in KBM™ Gold Basal Medium (Lonza; Cat#00192151; Walkersville, MD, USA) containing KGM™ Gold SingleQuots™ supplements (Lonza; Cat#:00192152). HEKs were incubated at 37 °C in a humidified atmosphere containing 5% CO_2_. HEKs between passages 6 and 8 were used for experiments and were collected using a trypsin-EDTA solution.

### 4.2. Cell Cytotoxicity Measurement

Cell cytotoxicity was measured using a Cell Counting Kit-8 (CCK-8) assay. Briefly, HEKs were seeded at a density of 1 × 10^4^ cells/well in 96-well plates overnight and then treated with different doses of baicalin (Sigma, St. Louis, MO, USA) at 37 °C for 24 h. After treatment, 10 μL of CCK-8 solution was added to culture medium followed by incubation for 1 h. The absorbance was measured at 450 nm using an ELISA microplate reader (VersaMax; Molecular Devices, California, CA, USA). Cell viability is presented as the percentage of the control.

### 4.3. Wounding Healing Assay

HEKs were seeded at a density of 5 × 10^5^ cells/well in 12-well cell culture plates. Cells were treated with baicalin. A scratch was made across the well on 100%-confluent cell monolayers using a 100 µL sterile micropipette tip, and the wells were washed with Dulbecco’s phosphate-buffered saline (Sigma) to remove debris and dislodged cells. Photographs of wound closure were taken at 0, 6, 24, and 48 h using an Olympus IX70 microscope equipped with a digital camera. Migration rates were calculated as the average percentage of the wound area using ImageJ software.
Wounding area (%) = (Wound area on day zero − Wound area on day A)/(Wound area on day zero) × 100
where A = days after wound induction.

### 4.4. Experimental Animals

Female BALB/c-nu nude mice, six weeks old and weighing 20 to 25 g, were purchased from the ORIENT BIO Animal Center (Seongnam, Seoul, Korea). All experimental procedures were conducted in accordance with the Animal Care and Use Committee (IACUC) at the Yonsei University College of Medicine (IAUAC_2020–0294). All mice were housed in cages at 23 ± 0.5 °C and 55–60% relative humidity with an automatically controlled light–dark cycle. Mice were allowed free access to food and water.

### 4.5. Animal Experimental Design

Mice were anesthetized with 1% isoflurane to insert an internal magnet. After disinfection with povidone-iodine, a 10 mm incision was made above the greater gluteus muscle [39,40,41]. A sterile magnet, 10 × 10 × 1 mm^3^, was placed above the gluteus muscle, and the incision site was sutured using biomedical silicon adhesive tape (SRTO; JH international, Incheon, Korea). Another magnet of the same size was placed directly above the implanted surface. We performed a previously described protocol [41], in which each cycle comprised 2 h of localized low pressure followed by release for at least 30 min. To induce the advanced-stage pressure ulcer model, we performed repeated cyclic application of the magnet and loaded pressure to induce ischemia–perfusion injury overnight and released for 1 day before evaluating this simulated clinical situation. We performed molecular and histological analyses to evaluate the established pressure ulcer model. Mice were randomly divided into two groups (*n* = 18)—a negative control group (PBS) and a baicalin group (10 mg/mL).

### 4.6. Histological Analysis

For histological analyses, skin specimens were harvested on days 1, 3, 7, and 10 post wounding. Tissue samples from each mouse were fixed in 10% formalin for more than 24 h and then processed using a routine paraffin embedding technique. After paraffin embedding, 4 µm thick sections were prepared. Sections were stained with H&E (ab245880; Abcam, Cambridge, UK) after deparaffinization and dehydration with alcohol solutions. The tissue sections were examined using a light microscope at different magnifications.

### 4.7. Quantitative Reverse Transcription-Polymerase Chain Reaction (qRT-PCR)

Mouse skin samples were homogenized using a TissueLyser II (Qiagen, Hilden, Germany). Total RNA was extracted using the RNAiso Plus kit (Takara Bio, Kusatsu, Shiga Prefecture, Japan) according to the manufacturer’s protocol and was quantified using a NanoDrop 2000 spectrophotometer (ThermoFisher, Waltham, MA, USA). After cDNA synthesis using the RNA to cDNA EcoDry™ premix kit (Takara Sake, Berkeley, CA, USA), mRNA levels were assessed using qRT-PCR and SYBR Green Master Mix (4309155; Promega Corporation, Madison, WI) on a QuantStudio 3 Real-Time PCR System (Applied Biosystems, Foster City, CA, USA). The cycling reaction conditions were as follows: 95 °C for 10 min, followed by 40 cycles of denaturation at 95 °C for 15 s, 60 °C for 20 s, and 72 °C for 30 s. The specific primer pairs used in the present study are listed in Appendix A. The mRNA levels were calculated using the relative quantification (ΔΔCt) method. β-actin was used as the housekeeping gene for normalization of gene expression levels.

### 4.8. Immunofluorescence Staining

For immunofluorescence staining, slide-mounted mouse skin cryosections (6 µm thick) were fixed using 4% paraformaldehyde (Cell Signaling Technology, Danvers, MA, USA) for 15 min at room temperature (RT). Slides were washed three times with phosphate-buffered saline (PBS)-T (PBS containing Triton X-100; DAEJUNG, Busan, Korea). Slides were incubated with primary antibodies overnight at 4 °C. After being washed three times, slides were incubated with a secondary antibody (Goat pAb to Rb IgG-FITC; ab6717; Abcam) for 1 h at RT. Next, sections were rinsed three times with PBS and DAPI (VECTASHIELD*^®^* mounting medium with DAPI, Vector Laboratories Inc., Burlingame, CA, USA) was applied for nuclear staining. Slides were scanned using a Zeiss microscope (LSM 700; Carl Zeiss, Jena, Germany) for image acquisition, and the fluorescence intensity was measured using ImageJ software (National Institutes of Health, Bethesda, MD, USA).

### 4.9. Statistical Analysis

All data were presented as means ± standard deviation (SD) from experiments performed at least in triplicate. Statistically significant differences were determined using two-tailed Student’s t-tests. Statistical analysis was performed using SPSS software version 25.0 (SPSS ver. 23; SPSS Inc., Chicago, IL, USA). A *p*-value < 0.05 was considered significant (* *p* < 0.05, ** *p* < 0.01, *** *p* < 0.005).

## 5. Conclusions

Baicalin administration promotes cutaneous wound healing of mouse pressure ulcers by suppressing the inflammatory response and inducing growth factors to promote epithelial migration and dermal reconstruction. Therefore, we suggest that baicalin could serve as a promising agent for the treatment of pressures ulcers.

## Figures and Tables

**Figure 1 ijms-24-00329-f001:**
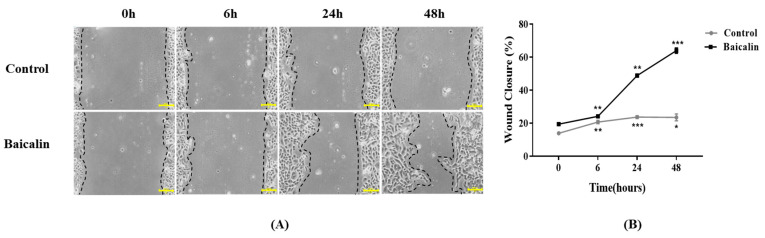
Baicalin enhances the rate of wound closure by human epidermal keratinocytes (HEKs). (**A**) Images of HEK scratch assays immediately after the scratch was made and then after 6, 24, and 48 h in the presence of baicalin. There was a significant increase in the extent of wound closure in HEK scratch assays in baicalin compared with that in the control medium at 48 h. (**B**) Statistical analysis of the wound closure area using ImageJ software. All data are expressed as mean ± SD from three independent experiments. ** p* < 0.05, *** p* < 0.01, *** *p* < 0.001. Scale bar: 200 μm.

**Figure 2 ijms-24-00329-f002:**
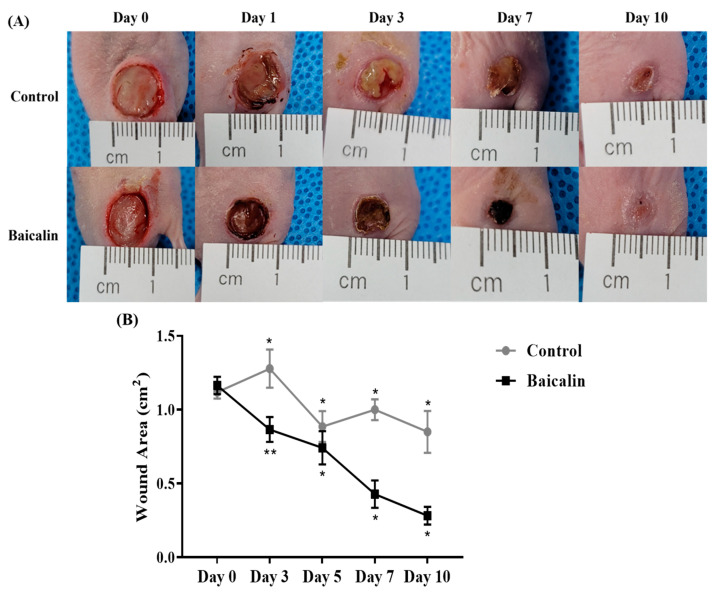
Topical application of baicalin accelerates the wound healing of pressure ulcers in mice. (**A**) The images show control and baicalin-treated wounds on days 0, 1, 3, 7, and 10 post-pressure ulcers in mice. (**B**) Statistical analysis of the wound area using ImageJ software. All images are representative of three independent experiments (*n* = 18). * *p* < 0.05, ** *p* < 0.01.

**Figure 3 ijms-24-00329-f003:**
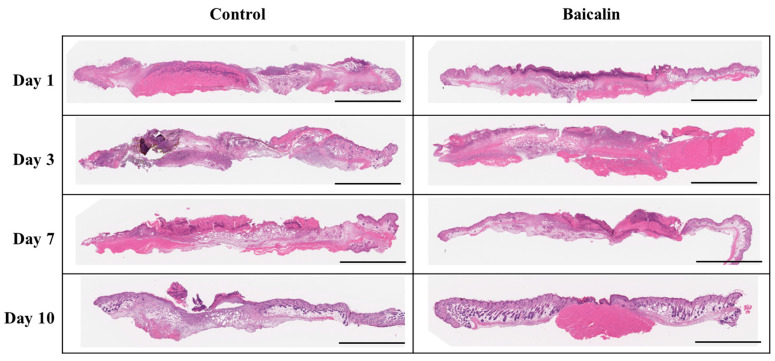
Histological analysis of the effect of baicalin on skin structure in murine pressure ulcers. Skin tissue samples from each group were collected on days 1, 3, 7, and 10 after treatment with baicalin and were stained with hematoxylin and eosin (H&E). All images are representative of three independent experiments (*n* = 18). Scale bar: 4 mm.

**Figure 4 ijms-24-00329-f004:**
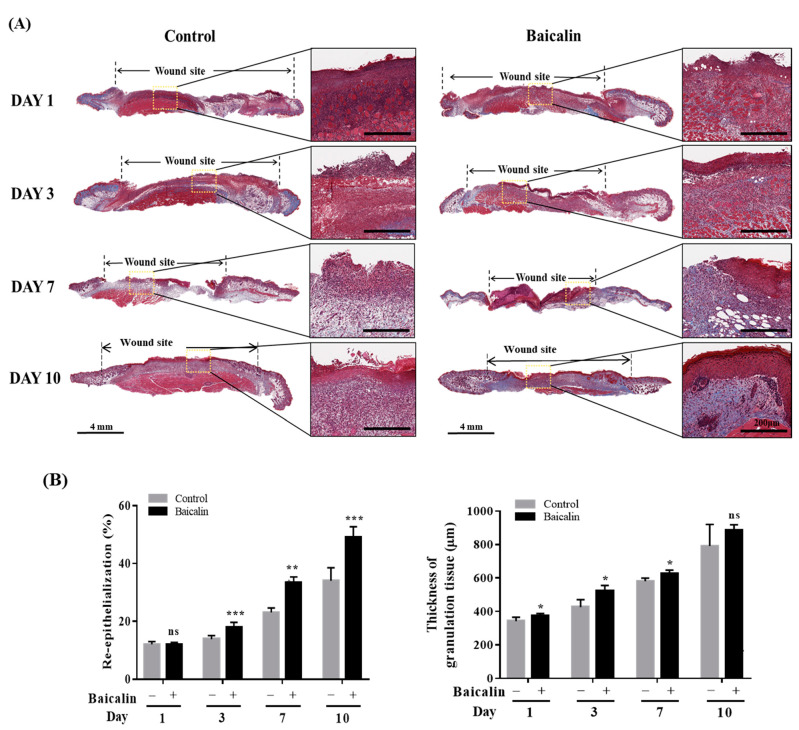
Treatment with baicalin accelerates the restoration of skin structure in a chronic murine model of skin ulcers. (**A**) Representative Masson’s trichrome (MT) images acquired at wound sites on days 1, 3, 7, and 10. Scale bar: 4 mm, 200 μm. (**B**) Quantitative analysis of the re-epithelialization and granulation tissue was performed using ImageJ. All images are representative of three independent experiments (*n* = 18). * *p* < 0.05, ** *p* < 0.01, *** *p* < 0.001, ns: not significant.

**Figure 5 ijms-24-00329-f005:**
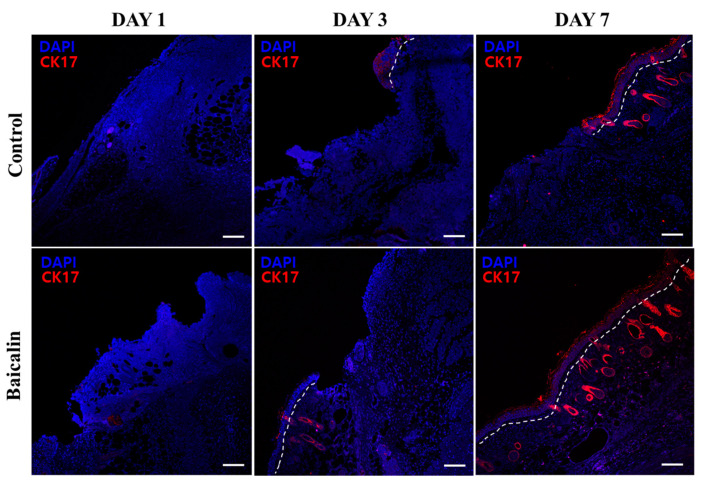
Baicalin promotes keratinocyte migration in pressure ulcer wound healing. Representative immunofluorescence images in histological sections of pressure ulcer skin lesions in mice. CK17 (red); DAPI (blue). The epithelial layer is distinguished from the dermis by a white dotted line. All images are representative of three independent experiments (*n* = 18). Scale bar: 100 μm. CK17, cytokeratin 17; DAPI, 4′,6-diamidino-2-phenylindole.

**Figure 6 ijms-24-00329-f006:**
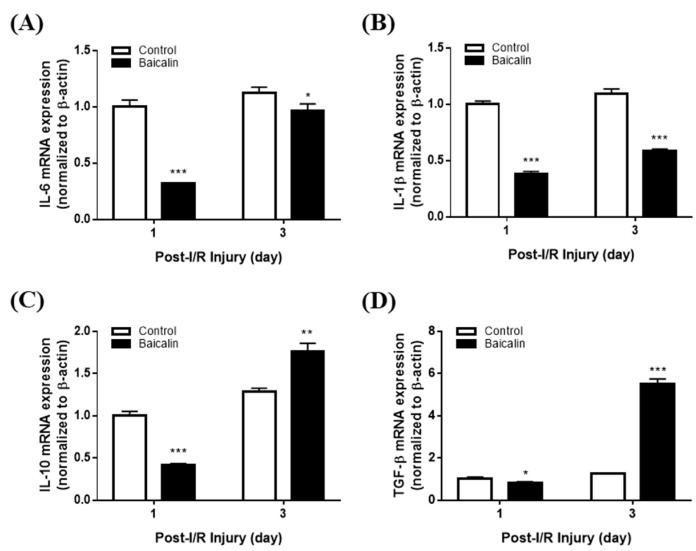
Relative mRNA expression levels of (**A**) IL-6, (**B**) IL-1β, (**C**) IL-10, and (**D**) TFG-β in mice with pressure ulcers. Baicalin suppresses pro-inflammatory cytokines and promotes anti-inflammatory mediators such as TGF-β in a pressure ulcer mouse model. All data are expressed as mean ± SD from three independent experiments. * *p* < 0.05, ** *p* < 0.01, *** *p* < 0.001. IL, interleukin; TGF-β, transforming growth factor-β.

**Figure 7 ijms-24-00329-f007:**
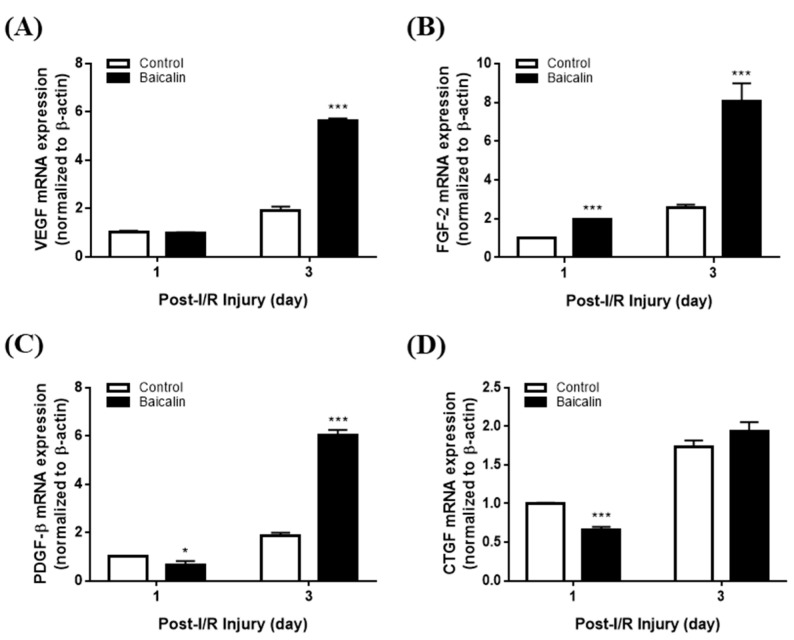
Gene expression levels of (**A**) VEGF, (**B**) FGF-2, (**C**) PDGF- β, and (**D**) CTGF, proangiogenic markers, in mice with pressure ulcers treated with baicalin. The expression levels of VEGF, FGF-2, and PDGF-β were significantly elevated in the baicalin group compared to the control group. All data are expressed as mean ± SD from three independent experiments. * *p* < 0.05, *** *p* < 0.001. VEGF, vascular endothelial growth factor; FGF-2, fibroblast growth factor-2; PDGF-β, platelet-derived growth factor- β; CTGF, connective tissue growth factor.

## Data Availability

Not applicable.

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
