# Peer review of "Effect of Baicalin on Wound Healing in a Mouse Model of Pressure Ulcers"

_ijms, 2022, doi:10.3390/ijms24010329_

Round 1
Reviewer 1 Report
The authors test the ability of Baicalin on healing of a pressure ulcer model. Overall, the studies appear fine but there are some issues that need to be resolved:
1) For many of the studies (figures 1, 2), there are statistical differences at day 0. What is your explanation for having significant differences at day 0? Does that influence your outcomes? You also have differences in most of the studies at day 1, which is typically in the "lag phase of healing", and one would not expect any differences. What is your explanation?
2) Your CK17 slides are not clear and appear to mainly show hair follicles. I would not expect that there would be hair regeneration in these wounds. So, what do these studies show?
3) Have you done any protein studies to confirm your mRNA results?
4) Are your studies repeatable? Have you done them more than once? What is the "N"?
5) You need to greatly expand your methods for the mouse model. I understand the pressure injury model but were all of the wounds open at the time of treatment? How did you apply the baicalin to the wounds? Topically? Systemically? How often was the treatment and how did you treat the wounds during the time of study? If the wounds were left open, how did you treat the wounds locally?
Reviewer 2 Report
Manuscript No.: ijms-2083360-peer-review-v1
Title: Effect of Baicalin on Wound Healing in a Mouse Model of Pressure Ulcers
IJMS
Reviewer's Decision: Accept after major revision
The authors of this research work describe the Baocalin effects on wound healing applications using a mouse model for pressure ulcers. The research is significant and should be published in the IJMS. However, the manuscript must be significantly improved before it can be published. As a result, I recommend accepting the manuscript after significant and satisfactory revisions. The following are the detailed comments:
Critical comments:
1. Title:
Seems to be ok
2. Abstract: The abstract is a comprehensive summary of the whole research article. The abstract contains numerous grammatical and formatting errors. It is suggested to improve the grammar and English language problems. The abstract section is more introductive and contains methodology information. It is suggested to reduce the informative and methodology section and add more results outputs with specific biomedical applications. The incomplete information in the abstract may confuse the readers.
3. Introduction: The introduction section discusses the drug delivery system, and it is preferable to start with the need for natural materials and their application in the drug delivery system. The introduction and the rest of the manuscript have several grammatical and formatting errors. Please improve the grammar, language, and formatting issues in the manuscript.
a. The page. 01, lines 34-37 "Pressure ulcers are a common skin disease that often occurs in older people and patients who are bedridden and have a physical inability. Pressure ulcers are caused by sustained pressure on localized tissue, leading to ischemia of the skin, subcutaneous tissue, and muscles surrounding the wound." should be written as "Pressure ulcers are a common skin condition that frequently affects elderly people and patients who are bedridden and physically incapable. Pressure ulcers are brought on by prolonged pressure applied to a specific area of tissue, which causes ischemia in the skin, subcutaneous tissue, and muscles close to the wound."
b. Page. 01, lines 41-44 "An early-stage pressure ulcer, because of its subtleness, is challenging for non-medical personnel to diagnose by visually identifying the lesion. Once the lesion progresses, epithelial injury and erosion at the site are evident, leading to ulceration of the repeatedly affected area." should be written as "Because of how subtle they are, early-stage pressure ulcers are difficult for non-medical staff to identify visually. The repeatedly affected area develops ulcers as the lesion progresses, causing epithelial injury and erosion at the site."
c. Page. 02, lines 50-54 "Wound healing is a complex process involving several overlapping stages and is controlled by the activity of cytokines and growth factors. In chronic wounds, the expression of inflammatory cytokines is upregulated and the wound microenvironment sequesters various growth factors and cytokines to inhibit their functions, thus resulting in delayed wound healing." should be written as "The activity of cytokines and growth factors regulates the complex process of wound healing, which has several overlapping stages. In chronic wounds, the expression of inflammatory cytokines is upregulated, and the wound microenvironment sequesters different growth factors and cytokines to inhibit their functions, delaying wound healing."
4. References: The manuscript lacks the literature citation of some highly interesting, most recent relevant works, and thus the reference citations are not up to date. Too many references have been given for a research article that may question the novelty of research work that may give an impression that so many people have reported the research already. These citations will help to explain cellular behavior, wound healing, and other biological behaviors for successful wound healing applications. In this regard, the author should refer to some of the most recent papers on hydrogel, such as
· Sajjad, A. et al. (2021). Development of antibacterial, degradable and ph-responsive chitosan/guar gum/polyvinyl alcohol blended hydrogels for wound dressing. Molecules, 26(19), 5937.
· Stojanović, et al. (2022). Multifunctional Arabinoxylan-functionalized-Graphene Oxide Based Composite Hydrogel for Skin Tissue Engineering. Frontiers in Bioengineering and Biotechnology, 10.
· Al-Arjan, et al. "pH-Responsive PVA/BC-f-GO Dressing Materials for Burn and Chronic Wound Healing with Curcumin Release Kinetics." Polymers 14, no. 10 (2022): 1949.
· Ghanadian, M., et al., (2022). The Effect of Plantago major Hydroalcoholic Extract on the Healing of Diabetic Foot and Pressure Ulcers: A Randomized Open-Label Controlled Clinical Trial. The International Journal of Lower Extremity Wounds, 15347346211070723.
· Shafiei, M., et al., (2021). A Comprehensive Review on the Applications of Exosomes and Liposomes in Regenerative Medicine and Tissue Engineering. Polymers, 13(15), 2529.
5. Materials and methods: This section is missing, and please add it; otherwise, it may confuse the readers.
· The author has used the same "degree" sign for temperature and angle. It is recommended to use the "degree" symbol accordingly throughout the manuscript.
6. Results and Discussions: The following issue must be taken into consideration.
a. The surface morphology is missing from the prepared sample to evaluate cell adherence.
b. Figure 1. It is recommended to use the appropriate color for scaling.
c. Use the proper font size for Figure 3 to make uniform figures as in Figures 1 and 2.
d. The figure caption and style should be identical throughout the manuscript to keep the continuity of the work.
7. Conclusions: The conclusion section is the most important summary of a research article, and it should be based on the conclusion for the conclusion. But it is missing, and it is recommended to provide the conclusion section to declare your research outcomes.
8. As per the comments given for the results and description.
In summary, the reported work has significant value; however, a major and thorough improvement/correction of language, grammar, syntax, etc., is necessary to improve the paper's quality and make it publishable in the IJMS.
· All the abbreviations should be defined before their 1st-time use.
Round 2
Reviewer 1 Report
The authors have addressed my concerns.
Reviewer 2 Report
All the comments have been addressed successful and the manuscript can be accepted in present form.